# Recent Advances in Psychopharmacology: From Bench to Bedside Novel Trends in Schizophrenia

**DOI:** 10.3390/jpm13030411

**Published:** 2023-02-25

**Authors:** Asim A. Shah, Syed Z. Iqbal

**Affiliations:** 1Barbara & Corbin J. Robertson Jr. Chair in Psychiatry, Menninger Department of Psychiatry, Division of Community Psychiatry, Family & Community Medicine, Baylor College of Medicine, Houston, TX 77030, USA; 2Chief of Psychiatry, Ben Taub Hospital/Harris Health System, Houston, TX 77030, USA; 3Menninger Department of Psychiatry & Behavioral Sciences, Baylor College of Medicine, Houston, TX 77030, USA

**Keywords:** molecular targets, psychopharmacology, molecular targets, GABA, pharmacogenomics, personalized medicine

## Abstract

Research in the field of psychopharmacology is ongoing to develop novel compounds which can revolutionize the treatment of psychiatric disorders. The concept of bench-to-bedside is a tedious process, transforming the initial research performed in the laboratories into novel treatment options. Schizophrenia (SCZ) is a chronic psychiatric illness with significant morbidity and mortality. SCZ not only presents with psychotic symptoms including hallucinations and delusions but also with negative and cognitive symptoms. The negative symptoms include the diminished ability to express emotions, loss of pleasure, and motivation with minimal social interactions. Conventional antipsychotics primarily target positive symptoms with minimal therapeutic benefits for negative and cognitive symptoms along with metabolic side effects. Researchers have explored novel targets to develop new compounds to overcome the above limitations. The glutamatergic system has provided new hope in treating schizophrenia by targeting negative and cognitive symptoms. Other receptor modulators, including serotonergic, phosphodiesterase, trans-amine-associated receptors, etc., are novel targets for developing new compounds. Future research is required in this field to explore novel compounds and establish their efficacy and safety for the treatment of schizophrenia. Last but not least, pharmacogenomics has effectively utilized genetic information to develop novel compounds by minimizing the risk of failure of the clinical trials and enhancing efficacy and safety.

## 1. Introduction

The history of psychopharmacology is divided into multiple phases, from the mid-nineteenth century to the 1950s, including the judicious use of drugs such as bromides, hyoscine, and paraldehyde, primarily for sedation and treating aggressive behavior [1]. Chlorpromazine was the first antipsychotic drug discovered by French Navy anesthesiologist Henry Laborit [2]. It was hypothesized that promethazine potentiates the anesthetic effects of other agents [2]. It was also postulated that antihistamines used during surgery could further relax the patient by lowering body temperature [3]. Courvoisier et al., in 1950, tested the compound chlorpromazine (4560RP) in rodents and determined it effective in prolonging sleep induced by barbiturates [2]. In 1951, Laborit and Huguenard tested a combination of chlorpromazine, promethazine, and an analgesic with a conventional anesthetic agent. They inferred that adding chlorpromazine required a lowered dose of the anesthetic agent with better tolerance of surgical trauma [2]. Chlorpromazine was initially marketed in 1952 as an antipsychotic by Rhone-Poulenc with the name of Largactil, “large CNS effect” [2]. This was followed by marketing in the US in 1954 by Smith Kline and French (Philadelphia, PA, USA) as Thorazine [2].

The advent of chlorpromazine started the era of modern psychopharmacology by shifting the paradigm from masking the symptoms to treating the disease process. The period of chlorpromazine flourished in the US, and by 1956, four million patients had already used chlorpromazine, amounting the profit to USD 75 million [1]. The era of chlorpromazine was followed by the discovery of other psychotropic medications, including meprobamate, chlordiazepoxide, monoamine oxidase inhibitor (MAOI) iproniazid, and tricyclic antidepressant imipramine [1]. These drugs were given various names, including major tranquilizers, ataractics, or neuroleptics, followed by the term antipsychotics until the late 1960s and 1970s [1]. Forty antipsychotic drugs were introduced worldwide by the 1990s, out of which 15 were in the US, including thioridazine, trifluoperazine, thiothixene, haloperidol, etc. [2]. The dopamine hypothesis has been widely accepted since the late 1970s and was initially proposed as the main etiological factor for schizophrenia [4].

To facilitate the development of novel compounds, randomized controlled trials (RCTs) were established in 1962. In the early 1970s, the Food and Drug Administration (FDA) made RCT the primary requirement to establish the safety and efficacy of novel compounds [1]. Developing a novel compound for treating mental illness is a tedious process requiring multiple steps. The first step in the research is identifying a new molecule that can be used for treatment. The new molecule undergoes pre-clinical research performed either in vitro or in vivo by using good laboratory practices and guidelines as outlined in the 21 Code of Federal Regulation (CFR) part 58.1 [5]. Preclinical studies help in identifying drug toxicity and efficacy in an animal model.

The preclinical research is followed by the clinical trials performed on volunteers or the diseased population. These clinical trials are divided into various phases. Phase 1 trial is conducted on a small scale, involving 20 to 100 healthy individuals or people with disease [5]. The goal of Phase 1 trials is to establish safety and effective dosage, and approximately 70% of novel compounds are estimated to move to the next phase of development [5]. Phase 2 clinical trial is conducted on a large scale, including the diseased population, further establishing the efficacy and safety of the medication. Approximately 33% of novel compounds are estimated to move to the next phase of development [5]. Phase 3 clinical trial recruits a larger patient population, further establishing the efficacy and safety of the novel compound with monitoring of treatment-emergent adverse effects. A total of 25% to 30% of the novel compounds move to the next development phase [5]. Phase 3 clinical trials are followed by Phase 4 trials which include post-marketing safety monitoring once the drug is approved by the FDA [5].

The continued use of psychotropics after their marketing identified further limitations in their efficacy as both typical and atypical antipsychotics primarily targeted the positive symptoms of schizophrenia, leaving the negative and cognitive symptoms untreated. These conventional antipsychotics are also associated with significant side effects, including metabolic side effects. The limitations in the efficacy of these conventional compounds generated questions in the mind of the researchers, including the following: what is next? Can alternate brain receptors be targeted to provide an effective cure? How to treat negative and cognitive symptoms of schizophrenia which contribute to significant morbidity? The researchers attempted to answer these questions by exploring novel receptor targets apart from dopamine, including the TAAR1 receptor agonist and serotonin HT1A receptor antagonist [3]. Research has discovered the role of acetylcholine receptors in the pathogenesis of schizophrenia which is the target of future studies. NMDA receptor modulators have been tested to counter the negative and cognitive symptoms of schizophrenia. These include biopterin glycine transport inhibitor (GLyT-1) luvadaxistat and glycine transport inhibitor BI425809 [3]. The more we understand the chemical neurotransmission and synaptic function, the further it generates more opportunities to develop novel compounds for treatment. The Accelerating Medicines Partnership for Schizophrenia (AMP–Schizophrenia), the Foundation for the National Institutes of Health (FNIH), and the Autism Biomarker Consortium for Clinical Trials (ABC-CT) are working to identify biomarkers that can further enhance the process of research and development for new treatment compounds [3]. In this article, we have summarized the novel compounds used for treating schizophrenia utilizing various receptor targets. We have also highlighted the role of pharmacogenomics in the research and development of new compounds.

## 2. Methodology

The manuscript writing of this review article was conducted by literature search using the abovementioned keywords to select relevant articles focusing on novel targets and treatment options for schizophrenia. We used PubMed, PsycINFO and EBSCO search engines to select the articles for review. The focus was on isolating the novel targets that are used to develop new compounds to treat schizophrenia. We mostly included recently published articles. We also cited information from various institutional websites, including FDA. The manuscript writing includes both the text elaboration of the literate with summarization in the form of tables. This is a scoping review.

## 3. Novel Treatment Options for Schizophrenia

Schizophrenia (SCZ) affects 21 million people worldwide, decreasing life expectancy by 15 years shorter compared to the general population [6]. The dopamine hypothesis is mainly accepted for the symptoms of SCZ, with increased dopamine release in the striatum resulting in positive symptoms. In contrast, decreased dopamine levels in the cerebral cortex result in negative and cognitive symptoms [6]. Other neurotransmitter systems involved in the pathogenesis of SCZ include the serotonergic, cholinergic, and glutaminergic systems [6]. The lack of effective treatment for negative and cognitive symptoms leads to the development of novel compounds which can target these symptoms with improved efficacy and tolerance. Various novel compounds have been introduced in the market or are in development. The dopaminergic receptors are further explored to develop new compounds with improved efficacy and safety. The novel compounds which predominately affect the dopaminergic receptors with some effect on the serotonergic receptors include Cariprazine, Brilaroxazine (RP5063), F17464, Lumateperone (ITI-007), Brexpiprazole and Lu AF35700 [7].

Cariprazine is a partial agonist at dopamine (D2) and D3 receptors with seven times greater affinity at the D3 receptors [6]. Its higher affinity for the D3 receptors makes it more effective in treating negative and cognitive symptoms of SCZ [6]. It is also a partial agonist at 5-HT1A and an antagonist at 5-HT2B receptors. Its affinity at 5-HT1A is one-fifth of the affinity to D2 receptors, whereas affinity at 5-HT2B is comparable to that of the D2 receptor [6]. Various studies have demonstrated the safety and efficacy of Cariprazine, leading to FDA approval in 2015 with the brand name Vraylar for treating SCZ and bipolar disorder. A supplemental new drug application is submitted to the FDA for approval as an adjunct treatment in major depressive disorder (MDD) [8]. Brexpiprazole is a partial dopamine agonist at D2 and D3 receptors with greater affinity for D2 receptors [6]. It also has greater affinity on D1 and D4 receptors than aripiprazole [6]. It has partial agonist activity at 5HT1A and antagonists at 5-HT2A and α1 receptors [6]. It was approved by the FDA in 2015 with the brand name Rexulti for treating SCZ and adjunct treatment of MDD [9]. Another compound with a higher affinity for D3 receptors under investigation is F-17464. It is a D3 receptor antagonist and a 5-HT1A partial agonist [10]. A phase 2 study has shown improvement in the Positive and Negative Syndrome Scale (PANSS) score with improved cognition using F-17464 compared to placebo [6]. The side effects noticed with this compound include insomnia, agitation, akathisia, and hyperprolactinemia [6]. Brilaroxazine (RP5063) is a partial agonist on D2, D3, and D4 receptors along with serotonin 5-HT1A and 5-HT2A receptors [6]. It is an antagonist of serotonin 5-HT2B, 5-HT2C, 5-HT6, and 5-HT7 receptors [6]. The half-life of this compound is 40 h, with steady-state concentration reaching eight days after initiation [6]. It has shown improvement in positive and cognitive symptoms of SCZ. This compound has established the safety and efficacy in Phase 2 double-blind, multicenter RCT, showing improvement in total PANSS score compared to placebo. The Brilaroxazine (RP5063) was compared in various doses, including 15 mg, 30 mg, and 50 mg, along with aripiprazole 15 mg as a placebo in patients with acute symptoms of SCZ [11]. This study showed significant improvement in PANSS score with all the doses mentioned above [11]. The improvement in symptoms with 15 mg of RP5063 was shown in 34% of patients, while 30 mg and 50 mg showed improvement in symptoms in 34% vs. 46%, respectively [11]. The difference in PANSS score was statistically significant with doses of 15 mg and 50 mg [11]. This compound also showed improvement in performance in trail-making A and B cognitive tests signifying benefits in cognitive symptoms resolution in SCZ [11]. Common side effects noted include insomnia and agitation. No weight gain or electrocardiogram EKG changes were noted in this study [11]. Phase 3 clinical trials with this compound are required to further assess its efficacy and safety in the acute and maintenance phase of SCZ [11].

Lu AF35700 is another investigational compound with antagonist activity at the D1, 5HT2A, and 5HT2 receptors [11]. Its safety and efficacy were studied in two clinical trials. A 16-week Phase 3 RCT compared the effectiveness of Lu AF35700 10 mg and 20 mg to Risperidone 4–6 mg and Olanzapine 15 to 20 mg [11]. Lu AF35700 did not show a statistically superior response compared to Risperidone and Olanzapine but exhibited an equally effective antipsychotic response and was well tolerated [11]. The common side effects reported were headache and weight gain at the higher dose of 20 mg [6]. Table 1 summarizes the novel compounds that predominantly modulate the dopaminergic system.

The novel compound that predominantly affects the serotonergic system for treatment of SCZ includes Roluperidone (MIN-101), Bifeprunox, AVN-211 (CD-008-0173), SB-773812, Idalopirdine (Lu-AE58054), Lumateperone (ITI-007) and Pimavanserin. Roluperidone (MIN-101) is a 5-HT2A and a sigma two receptor antagonist with some action as an alpha1-adrenergic antagonist, with low or no affinity for muscarinic, cholinergic, and histaminergic receptors. One Phase 2 trial did not show beneficial effects of Roluperiodne compared with placebo; however, another phase 2 trial showed improvement in PANSS total score and negative symptoms with a dose of 32 and 64 mg at 12 weeks. Patients on 64 mg also improved cognitive scores, especially in verbal memory [6]. A Phase 3 trial has completed the recruitment, and results are awaited to establish the safety and efficacy of this compound.

Lumateperone (ITI-007) is also tested for its antipsychotic properties. It has a 60 times higher affinity at 5-HT2A receptors as an antagonistic compared to D2 receptors [6]. It results in complete blockage of 5-HT2A receptors at the antipsychotic dose [6]. It is also a presynaptic D2 receptor agonist along with a postsynaptic D2 receptor blocker. It increases the phosphorylation of GluN2B glutaminergic receptors in the limbic system, thus enhancing the NMDA receptor’s activity [6]. It modulates serotonin and, to some extent, dopamine and glutamate neurotransmission and has a unique mechanism of action [24]. It is not only a novel treatment agent for SCZ but is also tested for use in behavioral symptoms in Alzheimer’s disease and bipolar disorder [24]. It was approved by the FDA with the brand name CAPLYTA for the treatment of SCZ in December 2019. Its safety and efficacy were established in two Phase 2 placebo-controlled RCT. In one Phase 2 clinical trial, the effectiveness of Lumateperone, 42 and 84 mg, was compared with Risperidone, 4 mg, and placebo. Patients with acute SCZ symptoms were randomized in a 1:1:1:1 ratio [25]. The change in the PANSS score from baseline between Lumateperone, 42 mg, compared to placebo was statistically significant *p* = 0.017, effect size = 0.42 [25]. A statistically significant effect was observed when Risperidone, 4 mg, was compared with placebo *p* = 0.013, effect size = 0.44 [25]. Lumateperone, 84 mg, did not show any significant change. None of the treatment groups showed a significant difference in PANSS negative score subsets [25]. The most common treatment-emergent adverse effects TEAE noted included somnolence 17% vs. 32.5% in Lumateperone, 42 mg vs. 84 mg, respectively [25]. The frequency of extrapyramidal side effects (EPS) with Lumateperone was 6.7%, which was not significantly different from the placebo at 6.3% [25]. The efficacy and safety of Lumateperone were also studied in a Phase 3 clinical trial when 24 mg and 42 mg of Lumateperone were compared with a placebo in acute symptoms of SCZ [25]. This study also demonstrated statistically significant effects as a change in PANSS total score at day 28 from baseline with Lumateperone 42 mg as compared to placebo *p* = 0.02, effect size −0.30 [25]. The side effect profile was similar to that of the Phase 2 trial. The total cholesterol, LDL, and prolactin levels decreased when patients were switched to Lumateperone from standard antipsychotic treatment [6]. Pimavanserin is approved for treating psychosis related to parkinsonism disease and is also evaluated for its efficacy in treating SCZ. It is a partial inverse agonist at the serotonin 5HT2A receptor [26]. It was initially approved in 2016 for the management of psychosis related to Parkinson’s disease, and then the decision was reviewed in 2018 and was upheld [6]. It has resulted in augmentation of the effect of Risperidone and Clozapine [26]. A large Phase 2 clinical trial did not show effects on PANSS total score but resulted in improved negative symptoms in stable patients with SCZ [6]. It does not cause extrapyramidal side effects but results in QTC prolongation in the elderly [6]. Table 2 summarizes the novel compounds that predominately modulate the serotonergic receptors in treating SCZ.

The phosphodiesterase inhibitors, including BI409306, phosphodiesterase 10A (PDE10A) inhibitors MK-8189, Roflumilast, and TAK-063, are investigated as novel compounds for the treatment of SCZ. TAK-063 has been studied for the treatment of SCZ. In the Phase 2 trial, the TAK-063 could not achieve the primary endpoint results. In contrast, a magnetic resonance imaging study has shown that TAK-063 has simulated ketamine-induced change in various brain parts [26]. MK8191 is a phosphodiesterase inhibitor 10A which modulates dopaminergic and glutaminergic transmission. Phase 2 trial did not show the efficacy of this compound. BI 409306 is a phosphodiesterase 9A (PDE9A) inhibitor that increases cGMP levels in the brain resulting in enhanced episodic and working memory in rodents [30]. It is tested to improve cognitive impairment in SCZ and Alzheimer’s disease [6]. In a Phase 2 trial, BI 409306 did not show significant improvement in cognition in SCZ. Future trials are required to establish the efficacy and safety of this compound. Table 3 summarizes the modulators of phosphodiesterase inhibitors.

The amino acid glycine, D-serine, D-cycloserine, and D-aspartate plays an essential role in modulating the activity of NMDR by acting as a co-activist [26]. Hypothetically, the use of these compounds help facilitates the functioning of NMDR, resulting in improved negative and cognitive symptoms in SCZ. Clinical trials have obtained mixed results with these compounds, especially with D-serine. The negative results were obtained from studies in which D-serine was used at a lower dose, whereas D-serine, when used at the dose of 60 mg/kg (4 gm/day), has demonstrated improvement in negative and cognitive symptoms [26]. The glycine reuptake inhibitors increase the amount of glycine in the synapses, enhancing the function of NMDR. Biopterin, a potent GlyT-1 inhibitor, has shown improvement in negative symptoms in a proof-of-concept study but failed to show a beneficial effect on cognition [26]. BI 425809 is a glycine transporter one inhibitor that enhances glutamatergic activity. Phase 1 trials in healthy male volunteers have shown increased glycine levels in cerebrospinal fluid (CSF). This increase in CSF levels was dose-dependent between 5–25 mg and peaked in 6 to 10 h [33]. The rise in glycine levels with the 50 mg dose was comparable to the 25 mg, but the 25 mg dose showed a zig-zag pattern of increased glycine levels in CSF [33]. Its efficacy was also evaluated with computer cognitive training in patients with SCZ [34]. The results of this Phase 2 trial are still pending. The statistically significant outcome in cognition was observed in another Phase 2 study using BI 425809 10 mg and 25 mg compared with placebo, with small effect sizes. FDA assigned BI 425809 a breakthrough therapy designation in May 2021 [26]. The drug has entered into Phase 3 trial, evaluating its safety and efficacy for improving cognitive symptoms in SCZ. The CONNEX program consists of three phases and three trials, including CONNEX-1, CONNEX-2, and CONNEX-3.

The Phase 3 clinical trial program CONNEX is designed to establish the safety and efficacy of BI 425809 for improving cognition in adults with SCZ. The program comprises three clinical trials, all of which are Phase 3 randomized, double-blind, placebo-controlled parallel group trials. The primary efficacy endpoint is the change from baseline in the overall composite T-score of the Measurement and Treatment Research to Improve Cognition in Schizophrenia (MATRICS) consensus cognitive battery (MCCB) [35,36,37]. The sponsor has planned to conduct an extension study to establish the long-term safety of this drug [38]. Glycine transporter one inhibitor is associated with a decrease in hemoglobin, resulting in anemia [6].

D-amino acid oxidase (DAAO) plays a role in the pathogenesis of SCZ. It helps the metabolism of D-serine, the co-agonist at NMDAR [39]. NMDR hypofunction is found in SCZ, with decreased D-serine levels in peripheral blood and CSF [39]. SCZ is also associated with increased levels of DAAO in the brain. Therefore, inhibiting DAAO activity results in increased levels of D-serine, enhancing the NMDAR function. DAAO antagonists, including sodium benzoate and luvadaxistat TAK-831, are promising novel compounds for treating SCZ [39]. BIIB-104 is an α-amino-3-hydroxy-5-methyl-4-isoxazole propionic acid (AMPA) receptor-positive allosteric modulator that affects synaptic plasticity [6]. A phase 2 double-blind multidose placebo-controlled trial is completed to establish the safety and efficacy of this compound, with the result still pending [40]. The positive allosteric modulator of NMDA receptors, including CAD-9303, is widely studied and is a potential target for future research and development.

The role of glutathione in modulating NMDR receptors is well known. Decreased levels of glutathione are observed in SCZ [26]. N-acetylcysteine (NAC) is a glutathione precursor and has a neuroprotective effect [26]. Research is ongoing to establish the effect of NAC in treating the cognitive symptoms of SCZ. NMDAR antagonists are also explored for treating negative and cognitive symptoms of SCZ. Memantine, an NMDA receptor antagonist approved by the FDA for treating moderate to severe Alzheimer’s, has produced mixed results when used as add-on in the treatment of SCZ [26]. At the same time, two metanalyses have shown a positive effect of memantine as an add-on to negative and cognitive symptoms in SCZ [26].

Noncompetitive NMDR antagonist deuterated (d6)-dextromethorphan hydrobromide and ultra-low-dose quinidine sulfate (AVP786) are currently being investigated for the treatment of negative symptoms of SCZ. The quinidine increases the level of dextromethorphan by inhibition of cytochrome P450 2D6. AVP 786 is FDA approved for treating agitation in Alzheimer’s disease [26]. The AMPA receptor modulation is also a potential target for treating SCZ and is under investigation, including CX516, CX614, CX691, LY451395, TAK137, and TAK-653 [26]. The mGluRs are also potential targets of novel compounds. LY2140023, which is GluR-2 PAM, did not show improvement in symptoms of SCZ [26]. Other GluR-2 PAM, including JNJ40411813/ADX71149 and AZD8529, are under investigation [26].

Evenamide (NW-3509) inhibits the release of glutamate in synapses by being a selective voltage-gated sodium channel blocker. It has decreased neuronal hyperexcitability in the prefrontal cortex and hippocampus [6]. It does not influence the monoamine pathways, mainly affected by conventional antipsychotic medications [41]. A phase 2 RCT has shown superior efficacy with add-on Evenamide to conventional antipsychotic treatment [6]. Table 4 summarizes the glutamatergic receptor modulators for the treatment of SCZ.

The trace amine-associated receptors (TAAR) are also novel targets that have shown promising results in treating SCZ. The compounds that target the TAAR1 receptors include Ulotaront SEP-363856 and Ralmitaront (RO6889450). The Ulotaront SEP-363856 is a new antipsychotic compound that acts distinctively as an agonist at TAAR1 [6]. Studies have shown that it has lower affinity at 5HT1A, 5-HT2A, and D2 receptors [6]. Animal studies have shown that Ulotaront therapeutic effects are independent of its effects on D2 receptors [6]. Ulotaront is in Phase 3 clinical trials and has received FDA breakthrough therapy designation for SCZ [50]. It has shown its efficacy and safety in large randomized, double-blind, placebo-controlled trials and six-month open-label extension study [50]. This study randomized patients with acute exacerbations of SCZ who either received Ulotaront or a placebo [6]. Significant improvement in the PANSS score was demonstrated in the Ulotaront group over four weeks [6]. After completing the four weeks, randomized trial patients entered 26 weeks of open-label trial. An additional change in PANSS score was observed in patients initially randomized to the Ulotaront group who then continued the 26 weeks of open-labeled trial. Similarly, a change in PANSS score −27.9 was also observed in groups who were started on Ulotaront in the open-label trial and initially received the placebo [6]. The long-term open-label study also showed improvement in cognitive symptoms. The main side effects reported by Ulotaront include sedation and agitation with less propensity for extrapyramidal side effects [6].

Remaltaront is a partial agonist at the TAAR 1 receptor. Two randomized, double-blind, placebo-controlled trials were conducted to establish the safety and efficacy of this compound. In one clinical trial, Remaltaront as monotherapy was compared to a placebo and also as an add-on therapy on antipsychotics. This trial is ongoing, and the expected study termination is May 2023 [51]. The other trial was terminated prematurely, as in the preliminary analysis, the primary endpoint was not reached [52]. Table 5 summarizes the TAAR modulators used in treating SCZ.

Long-acting injectables (LAI) for improving treatment adherence include Aripiprazole Lauroxil NanoCrystal^®^, and the first subcutaneous injectable LAI Perseris (RBP-7000) subcutaneous formulation of Risperidone was recently approved by FDA. Similarly, positive results were obtained for Risperidone ISM^®^, an LAI formulation of Risperdal administered monthly [7]. Paliperidone palmitate 6-month intramuscularly injectable and Risperidone subcutaneously injectable TV46000 are currently under investigation [7].

The metabolic side effects are a major concern with conventional antipsychotics, especially Quetiapine, Risperidone, Olanzapine, etc. The Samidorphan–Olanzapine combination counteracts the side effect of weight gain while maintaining Olanzapine’s efficacy [7]. It was recently approved by the FDA in 2021 with the brand name Lybalvi and has shown beneficial results in Phase 2 and 3 clinical trials. Table 6 summarizes the clinical trials conducted for the approval of Lybalvi.

Different other receptor complexes are involved in the exploration of novel compounds. Xanomeline is a muscarinic agonist acting on M1 and M4 types. It has minimal affinity for dopamine receptors D2 and D3 [59]. Its efficacy was established in a small clinical trial randomizing 20 patients with SCZ and schizoaffective disorder. It was determined to be effective in decreasing positive, negative, and cognitive symptoms of SCZ compared to placebo, although statistical tests for cognitive symptoms were not adjusted for multiple testing [6]. It was not associated with extrapyramidal and metabolic effects compared to other antipsychotics. It causes cholinergic side effects, including nausea (70%), vomiting (60%), gastrointestinal distress (70%), salivation (20%), diarrhea (20%), and constipation (20%) [6]. To enhance its tolerability, it was tested along with trospium, a peripheral muscarinic receptor antagonist [59]. This trial included n = 182 patients having acute exacerbations of SCZ. This study showed more significant improvement in PANSS scores, including the positive and negative scales in the Xanomeline–trospium group vs. placebo [59].

TAK-041 is an orphan G-protein-coupled receptor 139 (GPCR139) agonist [60]. GPCR 139 is highly expressed in the habenula, linked to depression, SCZ, and substance use disorder [60]. This compound is tested to assess its efficacy and safety for treating negative symptoms of SCZ. A phase 2 RCT of add-on TAK-041 to existing antipsychotic therapy was studied in 23 subjects. The primary endpoint was the improvement in the Brief Assessment of Cognition in Schizophrenia (BACS) and an assessment of blood oxygen level-dependent (BOLD) signals in the ventral striatum [61]. According to the summary of results, the study did not report a statistically significant improvement in cognitive function with TAK-041. [6,61]. The role of cannabidiol (CBD) is also investigated as a novel compound in treating SCZ. Studies have provided mixed results for using CBD in treating SCZ [26]. Dexmedetomidine is approved by FDA for acute agitation related to SCZ and bipolar disorder. Its safety and efficacy were established in randomized, placebo-controlled parallel group fixed-dose trials 1:1:1 ratio comparing Dexmedetomidine with a dose of 120–180 mcg and placebo [62]. Two studies were conducted, SERENITY-1, including n = 380 patients with SCZ with acute agitation, and SERENITY-2, including n = 378 agitated patients with bipolar I and II disorders. The primary endpoint was the change from baseline in the positive and negative syndrome scale excited component (PEC) two hours after initial dosing [62]. It binds to the alpha-2 receptors in Locus coeruleus to block the release of norepinephrine which is released in the state of hyperarousal. Sublingual Dexmedetomidine is effective in the management of agitation. Side effects observed in the studies include somnolence, oral paresthesia, dizziness, hypotension, orthostatic hypotension, and dry mouth [62].

The research for the development of novel compounds is still ongoing, with newer compounds under investigation. We hope that the treatment of SCZ in the future will not only focus on treating the positive symptoms but also provide remedies for the negative and cognitive symptoms with minimal side effects and improve the morbidity of the illness.

## 4. Role of Pharmacogenomics in the Development of Novel Compounds

Pharmaceutical companies experience significant challenges in the development of novel compounds. They must establish the safety and efficacy of the novel compounds at different levels in clinical trials. They are expected to generate three to four new chemical entities (NCE) yearly [63]. They have to face the failure of clinical research due to a lack of efficacy or safety. It is estimated that only 10% of novel compounds that enter clinical development are approved for marketing [63]. The cost of new drug development is substantial, as spending on research and development has tripled since 1990 to USD 26.4 billion [63]. The price per patient for the Phase 3 clinical trial for the CNS product is estimated to range from USD 8000 to USD 12,000 [63]. Very few psychopharmacological agents have entered the market in the last few decades. To minimize the waste of time, effort, and financial burden in developing novel compounds, pharmaceutical companies are utilizing a pharmacogenomic approach.

Pharmacogenomics and pharmacogenetics are used interchangeably. The pharmacogenomic includes genetic polymorphism, which can affect the individual’s ability to respond to medication, especially the difference in metabolism due to cytochrome P450 polymorphism resulting in adverse events [63]. The role of pharmacogenomics is to use genetic resources to enhance drug therapy with maximum efficacy and minimal adverse effects. The Pharmacogenomics Research Network (PGRN) consists of projects focusing on isolating genomic variants that enhance therapeutics and limit adverse drug reactions, leading to personalized medicine [64]. According to the 2016 Personalized Medicine Coalition (PMC), 20% of the novel compounds approved by the FDA are considered personalized medicines [64]. Pharmaceutical companies are willing to invest more resources in research on genetic biomarkers that can limit fatal side effects with novel compounds, avoiding legal and financial consequences. Due to this, they are conducting preliminary studies, including specific phenotype subgroups, ensuring that the efficacy and safety issues are identified in the early phase of a clinical trial [63]. Pharmacogenomics studies also help stratify the feasibility of continued development of a compound that has shown safety and efficacy only in a subset of patients with a particular phenotype. This limited efficacy and safety of a drug in a subpopulation has a risk of withdrawal from the market after approval due to serious adverse effects, e.g., Propulsid, Rezulin, Seldane, and Durant [63].

Similarly, the genetic traits that focus on efficacy based on receptor susceptibility will also be the target for developing effective novel compounds. Genetic sequencing can help researchers identify novel and effective targets and quickly introduce new compounds in the market for effective treatment. It also helps the researcher accurately interpret novel compound pharmacokinetics and pharmacodynamics [64]. In-depth knowledge of the human genome and its implication pharmacogenomics in drug development can substantially decrease the duration and cost of clinical trials by selecting the most responsive patients and enhancing the reliability of the drug development process.

## 5. Conclusions

Mental illnesses are prevalent throughout the world and significantly impact the overall quality of life of the individual. SCZ has significant morbidity and mortality. Initially, with the discovery of the typical antipsychotic targeting the dopamine receptors, the treatment aimed to eliminate the hallucinations and delusions named as the positive symptoms. This was followed by the discovery of atypical antipsychotics targeting the serotonergic and dopamine receptors. Subsequently, the clinicians realized the impact of the negative and cognitive symptoms on the patients’ quality of life. The existing treatment for SCZ helps to some extent with minimal impact on the negative and cognitive symptoms of schizophrenia. These limitations have compelled researchers to explore novel compounds with more efficacy and minimal treatment-emergent side effects.

Similarly, metabolic side effects are also a major problem with the use of antipsychotics resulting in discontinuation of treatment and non-adherence. The researchers have also focused on developing novel compounds that minimize this side effect. The concept of the bench-to-bedside in psychopharmacology is an extensive process that includes significant background research for discovering novel compounds and enormous financial resources to establish their safety and efficacy in the patient population via clinical trials. The novel compounds have to undergo the process of rigorous scrutiny before the FDA approves them for marketing. Even after marketing, the monitoring process continues by post-marking data analysis to further establish the safety of the novel compound. To provide adequate care, researchers are working diligently to discover new targets to provide effective treatment. Incorporating pharmacogenomics in developing novel compounds can further revolutionize the treatment of mental illness by ensuring the compound safety and efficacy. This can result in fast development, minimizing the chances of failed trials with cost-effectiveness. Continued efforts are required to accelerate the research and exploration of novel targets and develop new compounds for treating SCZ.

## Figures and Tables

**Table 1 jpm-13-00411-t001:** Modulators Predominantly Affecting Dopaminergic System.

Compound	Receptor Targets	Number	Dose	Control	Phase	Study Design	Primary Outcome	Results of Study
Cariprazine	Partial agonist D2 and D3, 5-HT1A, an antagonist at 5-HT2B, 5HT2A, H1, 5HT2C, α1 (low affinity)	330 (13–17 years)	1.5 mg, 4.5 mg	Placebo	Phase 3	Multicenter, randomized, double-blind, parallel-group, placebo-controlled study [12]	PANSS total score at week 6	Currently recruiting, estimated completion time April 2025
		732 with SCZ	Cariprazine 1.5 mg/d, 3.0 mg/d and 4.5 mg/d for 6 weeks	Placebo and Risperidone 4 mg/day	Phase 2	Multicenter, double-blind, randomized, placebo- and active-controlled, fixed-dose trial [13]	PANSS score at 6 weeks	↓ PANSS score in all three doses LSMD 1.5 mg/d, 3.0 mg/d 4.5 mg/d (−7.6, −8.8, −10.4), respectively; *p* < 0.001
		604 with SCZ	Cariprazine 3 mg/d and 6 mg/day	Placebo & Aripiprazole 10 mg/day	Phase 3	Multinational, randomized, double-blind, placebo- and active-controlled study [14]	PANSS score at 6 weeks	↓ PANSS score in 3 mg and 6 mg dose vs. placebo LSMD [95% CI]: 3 mg/d, −6.0 [−10.1 to −1.9], adjusted *p* = 0.0044; 6 mg/d, −8.8 [−12.9 to −4.7], adjusted *p* < 0.0001
		439 acute SCZ	Cariprazine 3 to 6 mg vs. 6 to 9 mg dose	Placebo only	Phase 3	Randomized, double-blind, placebo-controlled, parallel-group study [15]	PANSS score at 6 weeks	↓ PANSS score in both groups vs. placebo LSMD 3–6 mg/d: −6.8, *p* = 0.003; 6–9 mg/d: −9.9, *p* < 0.001)
Brexpiprazole	5-HT_1A_, D_2_, D_3_ partial agonist, 5-HT_2A_, 5-HT_2B_, 5-HT_7_, α_1A_, α_1B_, α_1D_, α_2C_ antagonist	674 SCZ	Brexpiprazole 1 mg, 2 mg, or 4 mg	Placebo only	Phase 3	Multicenter, double-blind, randomized, placebo-controlled trial of fixed-dose 1 mg, 2 mg and 4 mg Brexpiprazole [16]	PANSS score at 6 weeks	↓ PANSS score with 4 mg dose LSMD, −6.47; *p* = 0.0022, ↓ PANSS score was not statistically significant with 1 mg and 2 mg; LSMD −3.37, *p* = 0.1588; and −3.08, *p* = 0.1448, respectively
		636 SCZ	Brexpiprazole 0.25 mg, 2 mg, or 4 mg	Placebo only	Phase 3	Double-blind, randomized, placebo-controlled fixed-dose Brexpiprazole (0.25 mg, 2 mg, and 4 mg daily) compared to placebo [16]	PANSS score at 6 weeks	↓ PANSS score was statistically significant for the 2 mg and 4 mg doses of brexpiprazole (*p* < 0.0001 and *p* = 0.0006, respectively)
F-17464.	D3 antagonist 5-HT1A partial agonist	134 SCZ	F17464, 20 mg twice daily	Placebo only	Phase 2	Double-blind, randomized, placebo-controlled, parallel-group study [17]	PANSS score on day 43	↓ PANSS score was statistically significant F17464 vs. placebo *p* = 0.014
Brilaroxazine (RP5063)	Partial agonist D2, D3, D4, 5-HT1A and 5-HT2A antagonist 5-HT2B, 5-HT2C, 5-HT6 5-HT7	234 SCZ SAD	RP5063 15, 30, or 50 mg	Aripiprazole and placebo	Phase 2	Randomized, double-blind, placebo- and active-controlled randomized 3:3:3: 1:2 [18]	PANSS score on day 28	↓ PANSS score was statistically significant from baseline to day 28/EOT for the RP5063 15 mg and 50 mg arms (*p* = 0.0212 and *p* = 0.0167), the 30 mg arm did not reach statistical significance (*p* = 0.2733), although it was numerically superior
Lu AF35700	Antagonist at 5HT2A, 5HT2, D1	697 TRS	10 mg, 20 mg	Risperidone 4−6 mg, Olanzapine 15−20 mg	3	Double-blind randomized, parallel-group, active-controlled, fixed-dose ‘DayBreak’ [19]	PANSS at EOS 10 weeks	↓ PANSS score by 10% from randomization to EOT in all groups. No statistically significant differences between Lu AF35700 10 mg/day or 20 mg/day [95 % CI] in PANSS total scores were −0.12 [−2.37; 2.13] and 1.67 [−0.59; 3.94], respectively
		524 TRS	10 mg/day one week, then the dose can be increased to 20 mg or 70 mg once weekly	--------	-----	Open-label 52-week study with 6 weeks safety follow-up [19]	PANSS and TEAE	PANSS total scores showed a similar response pattern as for the double-blind study. LSMD of the change from baseline to week 52 score −8.45 [−10.08; −6.83]
		244 stable SCZ	32 mg/d, 64 mg/d	Placebo only	2b	Randomized, double-blind, placebo-controlled, parallel-group 12-week trials [20]	BACS	The MIN-101 32 mg group significantly improved the BACS verbal fluency *p* = 0.01, token motor *p* = 0.04, composite z scores *p* = 0.05) compared to the placebo
		156 completing 4 weeks of study	25 mg to 75 mg flexible dose	------	Ext of phase 2 trial------	Open-label 26-week study [21]	PANSS score at 26 weeks and TEAE	Response rates at week 26 (≥30% reduction in PANSS total score from the baseline of the double-blind study) (94.1% who received Ulotaront throughout and 92.5% for patients who initially received placebo in the double-blind study and were switched to Ulotaront in the extension study)
		435	50 mg or 75 mg in fixed dose	Placebo	3	A multicenter, randomized, double-blind, parallel-group, fixed-dosed 6-week study [22]	PANSS score 6 weeks	Results are awaited
		462	75 mg/d, 100 mg/d fixed dose	Placebo	3	Randomized, double-blind, parallel-group, placebo-controlled, multicenter [23]	PANSS score 6 weeks	Results are awaited

PANSS = Positive and negative syndrome scale, SCZ = schizophrenia, EOS = End of study, LSMD = Least square mean difference, TEAE = Treatment-emergent adverse effects, BACS = Brief assessment of cognition in schizophrenia ↓= Decrease.

**Table 2 jpm-13-00411-t002:** Modulators Predominantly Affecting Serotonergic Receptors.

Compound	Receptor Targets	Subjects	Dose	Control	Phase	Study Design	Primary Outcome	Results of Study
Roluperidone (MIN-101)	5-HT2A sigma 2 alpha1-adrenergic antagonist	244 SCZ	32 mg/d, 64 mg/d	Placebo only	2	Randomized, double-blind, placebo-controlled, multi-center trial [27]	PANSS negative score	MIN-101 32 mg/day and 64 mg/day groups showed↓ PANSS negative scores compared with the placebo group with effect sizes d = 0.45 and d = 0.57, respectively
		244 stable SCZ	32 mg/d, 64 mg/d	Placebo only	2b	Randomized, double-blind, placebo-controlled, parallel-group 12-week trials [20]	BACS	MIN-101 32 mg group showed significant improvement in BACS verbal fluency *p* = 0.01 and token motor *p* = 0.04 composite z scores *p* = 0.05 compared to the placebo group
Bifeprunox	Partial agonist at presynaptic 5-HT1ARs and D2Rs	93 acute SCZ	30 or 40 mg/day	Risperidone 4 to 6 mg/day	3	Randomized double-blind RCT [11]	PANSS total scores	Bifeprunox (30 or 40 mg/day) is less effective than Risperidone (4 or 6 mg/day) at all time points
		223 stable SCZ	20 mg/day	Quetiapine (600 mg/day)	3	Randomized double-blind RCT [11]	PANSS total scores	PANSS total score improved without any changes in functional outcome and quality of life
AVN-211 (CD-008-0173)	5HT6R antagonist	42	4 mg of AVN-211 (CD-008-0173)	Placebo	2	Randomized, double-blind, placebo-controlled, add-on, 4-week trial [28]	PANSS total scores	(PANSS) score improved in the treatment group *p* = 0.058
SB-773812	5HT2AR, D2R, D3R antagonists	317 acute SCZ	60, 120 mg	Olanzapine	2	12-week, multi-center, double-blind RCT [11]	PANSS total score	Statistically significant efficacy with both 60 and 120 mg in reducing PANSS score
Idalopirdine (Lu-AE58054)	5HT-6R antagonists	122 SCZ	120 mg/day augmentation on Risperidone	Placebo	2	12-week, double-blind, parallel-group, fixed-dose RCT [11]	PANSS total score	Augmentation therapy with Idalopirdine was not superior to placebo
Lumateperone (ITI-007)	Antagonist 5-HT2A, D2 receptors	335 acute SCZ	Lumateperone base, 42 and 84 mg	Placebo and Risperidone, 4 mg	2	Randomized, double-blind, placebo-controlled, multi-center trial 1:1:1:1 [25]	PANSS day 28	↓ PANSS score LSMD −7.4 ± 1.68 for placebo, −13.2 ± 1.69 for 42 mg, −8.3 ± 1.69 for 84 mg, and −13.4 ± 1.72 for Risperidone, 4 mg The difference between placebo to 42 mg (*p* = 0.017; effect size = 0.42) and Risperidone, 4 mg (*p* = 0.013; effect size = 0.44), were statistically significant
		450 acute SCZ	lumateperone base 24 and 42 mg.	Placebo only	3	Randomized, double-blind, placebo-controlled, multi-center trial 1:1:1: [25]	PANSS day 28	↓ PANSS score statistically significant with 42 mg when compared with placebo LSMD −4.2 (95% CI = −7.8 to −0.6, *p* = 0.02, effect size = −0.30); 24 mg failed to separate from placebo in efficacy
		301 SCZ stable	Safety &efficacy of lumateperone 42 mg	----	----	Open-label outpatient 6-week study (ITI-007-303) [29]	PANSS day 42 TEAE	↓ PANSS score with 42 mg significantly improved (unadjusted *p* < 0.001) relative to the previous antipsychotics baseline (mean change from baseline −2.2, [95% CI, −3.2, −1.2]

PANSS = Positive and negative syndrome scale, SCZ = Schizophrenia, LSMD = Least square mean difference, d = Cohen effect size. ↓= Decrease.

**Table 3 jpm-13-00411-t003:** Modulators of Phosphodiesterase.

Compound	Receptor Targets	Subjects	Dose	Control	Phase	Study Design	Primary Outcome	Results of Study
BI 409306	9A (PDE9A) inhibitor	450	10, 25, 50, or 100 mg	Placebo	2	The double-blind, parallel-group 12-week trial with randomization sequence as 2:1:1:1:1 [31]	CANTAB at 12 weeks, Stage 1	No difference in CANTAB score between BI 409306 and placebo in *n* = 120 subjects; the primary endpoint of improvement in cognitive function was not established; BI 409306 was well-tolerated, with an acceptable safety profile
							MCCB at week 12, Stage 2	There was no significant difference between BI 409306 and placebo in MCCB composite score change
MK-8189	phosphodiesterase 10A inhibitor	576	6 mg, 24 mg	Placebo	2B	Randomized, double-blind, placebo- and active-controlled 12-week trial	PANSS score 6 weeks, number of patients ≥ 1, TEAE up to 14 weeks; number of patients dropouts due to TEAE—12 weeks	Recruiting patient completion date: June 2024
Roflumilast	phosphodiesterase type-4 inhibitor	18	100 µg and 250 µg add on to SGA	Placebo	1	Randomized, double-blind, placebo-controlled, crossover design study [32]	Auditory steady-state response (early stage), mismatch negativity, and theta (intermediate stage) and P300 (late stage) was examined using an electroencephalogram	Roflumilast, 250 µg, significantly enhanced the theta oscillations amplitude related to mismatch negativity (*p* = 0.04) and working memory (*p* = 0.02) compared to placebo but not in any other cognitive markers

CANTAB = Cambridge Neuropsychological Test Automated Battery (CANTAB), MCCB = MATRICS Consensus Cognitive Battery, PT = Patients, TEAE = Treatment-emergent adverse effect, SGA = Second generation antipsychotic.

**Table 4 jpm-13-00411-t004:** Modulators of Glutamatergic Neurotransmission.

Compound	Receptor Targets	Number of Subjects	Dose	Control	Phase	Study Design	Primary Outcome	Results of Study
BI 425809	Activation of NMDAR by glycine GlyT1 inhibition	25 healthy male volunteers	5, 10, 25, or 50 mg, 6 subjects per group	------	1	Non-randomized, open-label, sequential-group, multiple-dose phase [33]	Glycine levels in CSF	50% increase in CSF glycine levels with 10 mg with a max level in 6 to 10 h. The increase was dose-dependent, with effects similar to those of 25 and 50 mg doses
		509	2 mg, 5 mg, 10 mg, or 25 mg	Placebo	2	Randomized, double-blind, placebo-controlled, parallel-group trial (1:1:1:1:2) for 12 weeks [42]	MCCB at 12 weeks	Improved cognition at 12 weeks with BI 425809 AMD with 10 mg vs. 25 mg 1.98, 95% CI 0.43–3.53 d = 0.34 vs. 1.73 [0.18–3.28] d = 0.30
		200	BI 425809	Placebo	2	Randomized, double-blind, placebo-controlled parallel group trial [34]	MCCB week 12	Results awaited, study completion date November 2022
		586 in each trial	BI 425809	Placebo	3	Randomized, double-blind, placebo-controlled parallel group 26-week trial	MCCB week 26	CONNEX-1 [35], CONNEX-2 [36], CONNEX -3 [37]; actively recruiting patients; completion date May 2024
		1401	BI 425809	------	3	An open-label, single-arm extension trial of CONNEX	TEAE	To establish long-term safety of BI425809 for one year [38]; estimated completion date: May 2025
Sodium Benzoate (SB)	Modulation of NMDR by DAAO antagonist	52	1 gm add-on treatment	Placebo		Double-blind, placebo-controlled 6-week randomized trial in two major medical centers in Taiwan [43]	PANSS total score; MCCB	Improvement in PANSS total, positive, and negative score in SB vs. placebo (*p* < 0.001); d = 1.16–1.69; improvement in neurocognitive composite score after treatment with SB *p* = 0.04, d = 0.67); SB was effective in the speed of processing (*p* = 0.03, d = 0.65), visual learning and memory (*p* = 0.02, d = 0.70)
		63	Sarcosine (2 g/day) or Sarcosine (2 g/ + Benzoate 1 g/day as add-on treatment	Placebo		Double-blind, randomized, placebo-controlled 12-week trial; 1:1:1 ratio [44]	PANSS total score; MCCB	SAR + SB effectively improved the GAF [SE] 0.16 ± 0.06, *p* = 0.005 after 12 weeks of treatment; however, there was no significant group difference in improvement in the PANSS and CGI-S scores; SB improved global composite cognition [SE] by 7.77 ± 2.74, *p* = 0.007, and neurocognitive composite score [SE] by 8.41 ± 3.07, *p* = 0.009. SAR + SB was modestly superior in improving global composite cognition compared to the SAR group alone
		60	1 gm, 2 gm as add-on Clozapine in treatment-resistant schizophrenia	Placebo		Double-blind, placebo-controlled 6-week randomized trial; inpatient study [45]	PANSS total score; QOLS GAF, SANS	SB group showed improvement in SANS scores vs. placebo (mean differences from baseline to week six SB 1 gm = 5.8 ± 7.5, SB 2 gm = 5.0 ± 6.5, placebo = 1.7 ± 2.8) improvement in PANSS score (mean differences from baseline to week 6 were SB 2 gm = 6.9 ± 4.7, placebo = 3.6 ± 3.2)
		100	500 mg of BZ twice daily as an add-on in patients with early psychosis aged 15–45 years	Placebo		Randomized clinical 12-week multi-center trial of either BZ at 1 g/d or placebo as addon to antipsychotic treatment [46]	PANSS total score at 12 weeks	The BZ group did not show improvement in total PANSS score compared to placebo with LSMD = −1.2 (2.4) *p* = 0.63; the study did not show any benefit of a twice daily use of BZ, 500 mg, in individuals with early psychosis
TAK-831 luvadaxistat (NBI-1065844	indirect modulation of NMDR by DAAO antagonist	256	Luvadaxistat as add on antipsychotic	Placebo		Randomized, double-blind, placebo-controlled, parallel-group 12-week multi-center study (interact study) [47]	PANSS negative symptoms score at day 84	The study did not meet the primary efficacy endpoint; the secondary endpoints of cognitive improvement were met but require further clinical evaluation
		308	Luvadaxistat as add on antipsychotic	Placebo		Randomized, double-blind, placebo-controlled, parallel-group study in subjects with cognitive impairment in schizophrenia, followed by open-label treatment erudite [48]	BAC composite score; baseline to day 98	Ongoing; actively recruiting patients
CAD9303	positive allosteric modulator (PAM) (NMDA) receptors	103	1	3 mg up to 1000 mg total daily dose	Placebo	Evaluate the safety, tolerability, pharmacokinetics, and effects on neurophysiological biomarkers of CAD-9303 oral treatment in subjects with schizophrenia and normal healthy volunteers [49]	Safety and tolerability of CAD-9303 day 1 through day 7 follow-up incidence of adverse events	Results awaited; study completed in December 2021
BIIB-104	195 SCZ	(AMPA) receptor-positive allosteric modulator	2	0.5 mg twice a day, orally, for 12 weeks	Placebo	Randomized, double-blind, multiple-dose, placebo-controlled study for CIAS	(MCCB) Working memory domain score to week 12	Study completed, but results are pending

CANTAB = Cambridge Neuropsychological Test Automated Battery (CANTAB), MCCB = MATRICS Consensus Cognitive Battery, PT = Patients, TEAE = Treatment-emergent adverse effect, SGA = Second generation antipsychotic, AMD = Adjusted mean difference, d = Effect size, SB = Sodium benzoate, SAR = Sarcosine, GAF Global Assessment of Functioning, SE = Standard Error, CGI-S = Clinical global impression of severity of illness, QOLS = Quality of life scale, SANS = Scale for the assessment of negative symptoms, SMD = Square mean difference, CIAS = Cognitive impairment associated with schizophrenia, BAC = Brief Assessment of Cognition in Schizophrenia.

**Table 5 jpm-13-00411-t005:** Modulators of TAAR Receptors.

Compound	Receptor Targets	Subjects	Dose	Control	Phase	Study Design	Primary Outcome	Results of Study
Ulotaront SEP-363856	agonist activity at (TAAR1)	245	50 mg or 75 mg fixed dose	Placebo	2	Randomized, double-blind, placebo-controlled, 4-week study; 1:1 ratio [53]	PANSS at four weeks	The mean change in PANSS score between Ulotaront and placebo was −17.2 vs. −9.7 points, respectively (LSMD −7.5 95% CI, −11.9 to −3.0; *p* = 0.001).
		156 completing four weeks of study	25 mg to 75 mg flexible dose	------	Ext of Phase 2 trial------	Open-label 26-week study [21]	PANSS score at 26 weeks and TEAE	≥30% reduction in PANSS total score from the baseline of the double-blind study. A total of 94.1% patients received Ulotaront throughout, and 92.5% of patients who initially received a placebo in the double-blind study were switched to Ulotaront in the extension study. A total 67% completion rate at week 26.
		435	50 mg or 75 mg in fixed-dose	Placebo	3	Multi-center, randomized, double-blind, parallel-group, fixed-dosed 6-week study [22]	PANSS score 6 weeks	Actively recruiting with an estimated completion date of January 2023
		462	75 mg/d, 100 mg/d fixed dose	Placebo	3	Randomized, double-blind, parallel-group, placebo-controlled, multi-center [23]	PANSS score 6 weeks	Actively recruiting with an estimated completion date of February 2023
		300 SCZ clinically stable	50 to 100 mg/day	Quetiapine XR (400 to 800 mg/day)	3	Randomized, double-blind, active comparator-controlled 56-week study to evaluate the long-term safety and tolerability of SEP-3638 [54]	PANSS and TEAE	Results are awaited
		586	25 mg, 50 mg	Placebo	3	Randomized, double-blind, placebo-controlled parallel group 28-week trial [35] (CONNEX-1)	MCCB after 26 weeks	Still recruiting patients; May 2024—expected completion date
		586	BI 425809	Placebo	3	Randomized, double-blinded, placebo-controlled parallel group 26-week trial (CONNEX-2) [36]	MCCB after 26 weeks	Still recruiting patients; May 2024—expected completion date
		586	BI 425809	Placebo	3	Randomized, double-blind, placebo-controlled parallel group 26-week trial (CONNEX-3) [37]	MCCB after 26 weeks	Recruiting patients; May 2024—expected completion date
Ralmitaront (RO6889450)	TAAR1 receptor partial agonist	247 SCZ, SAD	Oral dose of RO6889450 once daily	Placebo	2	Randomized, doublebBlind, placebo-controlled study [51]	BNSS Avolition/Apathy at 12 weeks	Recruiting patients; May 2024—expected completion date
		286 SCZ, SAD	150 mg once daily (QD)	4 mg QD Risperidone placebo	2	Randomized, double-blind, parallel group, placebo-controlled trial [52]	(PANSS) total score at week four.	The preliminary analysis did not achieve the primary endpoint, due to which the study was discontinued

PANSS = Positive and negative syndrome scale, SCZ = schizophrenia, SAD = Schizoaffective disorder, LSMD = least square mean difference, d = Cohen effect size, BNSS = Brief Negative Symptoms Scale, TEAE = Treatment Emergent Adverse events, MCCB = MATRICS Consensus Cognitive Battery.

**Table 6 jpm-13-00411-t006:** Samidorphan and Olanzapine combination.

Study	N	Age	Phase	Dose	Type of Study	Primary Endpoint	Results
ALK3831-302	309	18–50, BMI 17 to 30	2	OlZ + placebo, OlZ + 5 mg SAM, OLZ +10 mg SAM, OLZ + 20 mg SAM; 1:1:1:1	Proof-of-concept, randomized, controlled trial (RCT) Part A: 12-week double-blind Olanzapine-controlled treatment period; Part B: 12-week active-controlled treatment, dose-blinded	PANSS at week 12	ALKS 3831: all doses resulted in PANSS score reductions comparable to Olanzapine (mean difference ALKS 3831 vs. Olanzapine: 0.6 points, 95% confidence interval (CI): −1.2–2.5). A total 37% lower mean weight gain in ALKS 3831 compared to Olanzapine at 12 weeks, *p* = 0.006; 51% lower mean weight gain in ALKS 3831 vs. Olanzapine in a subset of patients who gain weight early in treatment, *p* ≤ 0.001 [55]
ENLIGHTEN-1 Study ALK3831-A305	403	18–70, BMI 18–40 kg/m^2^	3	Olz 10 mg, 20 mg and placebo Olz + SAM 10 mg/10 mg or 20 mg/10 mg	Double-blind, placebo- and active-controlled 4-week, randomized study [56]	PANSS total score at week 4	Treatment with OLZ/SAM significantly improved PANSS score LSMD ± SE: −6.4 ± 1.8 *p* ≤ 0.001; Olanzapine alone showed similar improvement; LS mean ± SE: −5.3 ± 1.84 *p* = 0.004
ENLIGHTEN-2 Study ALK3831-A303	561	18–55, BMI18–30 kg/m^2^	3	OlZ 10 mg + Sam 10 mg or OlZ 20 mg SAM 20 mg; dose flexible for the first 4 weeks and then fixed	Double-blind, active-controlled 24-week, randomized study to assess weight gain [57]	Change from baseline in bodyweight with a proportion of patients with ≥10% weight gain at week 24	LSMD % change in weight from baseline in Olz + Sam group 4.2 SE:0.7 was statistically significant vs. OlZ group 6.6 SE:0.7; more than 10% weight gain at 24 weeks was observed in 17.8% in the Olz + Sam group vs. 29.8% in Olz group alone
ALK3831-304	266	18–55	3	OLZ/SAM was administered at daily doses of 10 mg/10 mg (10/10), 15 mg/10 mg (15/10), or 20/10 mg	Open-label extension treatment of enlighten 2: ALKS 3831 52-week study [58]	Safety and tolerability	OLZ/SAM was well tolerated over 52 weeks of treatment; weight, waist circumference, and metabolic parameters were stable over 52 weeks of treatment; the long-term durability of OLZ/SAM treatment, as evidenced by sustained improvements in symptoms of schizophrenia over 52 weeks
ALK3831-A306	277	18–55	3	OLZ/SAM was administered at daily doses of 10 mg/10 mg (10/10), 15 mg/10 mg (15/10), or 20/10 mg	Open-label extension treatment arm of enlighten 1 study: ALKS 3831 52-week study	Safety and tolerability	Improvement in schizophrenia symptoms at 52 weeks with OLZ/SAM use; mean increases in weight were stabilized by week 6 with little further change through the end of treatment

OLZ = Olanzapine, SAM = samadorphin, SE = standard Error.

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
