# Peer review of "Recent Advances in Psychopharmacology: From Bench to Bedside Novel Trends in Schizophrenia"

_jpm, 2023, doi:10.3390/jpm13030411_

Round 1
Reviewer 1 Report
The manuscript deals with an important topic that is of interest to the audience of the journal. Reviewer has the following comments to make-
1. Based on the category of the manuscript (type of review), the inclusion of methodology is suggested to make sure that the review process was adequate and appropriate.
2. The table presented nicely depicted the novel approach. However, reviewers would suggest a table to present the unique characteristics of the group of medications (apart from receptor profiles) based on available limited data.
Author Response
Methodology part added as recommended ,
The unique charecteristics elaboration will be beyound the scope of this review as most of the noval compunds are still in developement process, and this can be done once they are fully approved by the FDA
Reviewer 2 Report
Very well done review. I only suggest to clearly report that this is a scoping review.
Author Response
The ammendment done as recommended by the reviewer